# Intrinsic Social Motivation via Causal Influence in Multi-Agent RL

## Abstract

We derive a new intrinsic social motivation for multi-agent reinforcement learning (MARL), in which agents are rewarded for having causal influence over another agent's actions. Causal influence is assessed using counterfactual reasoning. The reward does not depend on observing another agent's reward function, and is thus a more realistic approach to MARL than taken in previous work. We show that the causal influence reward is related to maximizing the mutual information between agents' actions. We test the approach in challenging social dilemma environments, where it consistently leads to enhanced cooperation between agents and higher collective reward. Moreover, we find that rewarding influence can lead agents to develop emergent communication protocols. We therefore employ influence to train agents to use an explicit communication channel, and find that it leads to more effective communication and higher collective reward. Finally, we show that influence can be computed by equipping each agent with an internal model that predicts the actions of other agents. This allows the social influence reward to be computed without the use of a centralised controller, and as such represents a significantly more general and scalable inductive bias for MARL with independent agents.

## 1 Introduction

Deep reinforcement learning (RL) has made impressive progress on specific tasks with well-defined reward functions, but is still difficult to learn intelligent behavior that generalizes across multiple domains. *Intrinsic motivation* is a technique for solving this problem by developing general reward functions that encourage an agent to learn across a variety of tasks (Singh et al., 2004). Previous approaches to intrinsic motivation have broadly fallen into two categories: (1) curiosity, or a drive for novelty (e.g. Pathak et al. (2017); Schmidhuber (2010)), and (2) empowerment, or a drive to be able to manipulate the environment (Klyubin et al., 2005).

We posit that this body of work has largely overlooked an important intrinsic motivation that is key to human learning: social interaction. Humans have remarkable social learning abilities; some authors suggest that it is social learning that has given rise to cultural evolution, and allowed us to achieve unprecedented progress and coordination on a massive scale (van Schaik & Burkart, 2011; Herrmann et al., 2007). Others emphasize that our impressive capacity to learn from others far surpasses that of other animals, apes, and even other proto-human species (Henrich, 2015; Harari, 2014; Laland, 2017).

Therefore, we propose an intrinsic reward function designed for multi-agent RL (MARL), which awards agents for having a causal influence on other agents' actions. Causal influence is assessed using counterfactual reasoning; at each timestep, an agent simulates alternate, counterfactual actions that it could have taken, and assesses their effect on another agent's behavior. Actions that lead to relatively higher change in the other agent are considered to be highly influential and are rewarded. We show how this reward is related to maximizing the mutual information between agents' actions, and is thus a form of social empowerment. We hypothesize that rewarding influence may therefore encourage cooperation between agents. We also take inspiration from experiments in human cognition, showing that newborn infants are sensitive to correspondences between their own actions and the actions of other people, and use this to coordinate their behavior with others (Tomasello, 2009; Melis & Semmann, 2010).

To study our proposed social influence reward in the MARL setting, we adopt the Sequential Social Dilemmas (SSDs) of Leibo et al. (2017). These are challenging MA environments with a game-theoretic reward structure, similar to Prisoner's Dilemma. For each individual agent, 'defecting' (non-cooperative behavior) has the highest payoff. However, the collective reward will be better if all agents choose to cooperate. The paradoxical payoff structure of these tasks make achieving cooperative social dynamics

extremely challenging for typical RL agents. We show that social influence allows agents to learn to cooperate in these environments, and make the following contributions:

- We demonstrate that deep RL agents trained with the proposed social influence reward cooperate to attain higher collective reward than baseline deep RL agents (Mnih et al., 2016). In some cases, this cooperation is attained because influencer agents learn to use their actions as an emergent communication protocol, analogous to behavior seen in animals (von Frisch, 1969).

- Motivated by the previous point, we apply the influence reward to training deep RL agents to use an explicit communication channel, as in Foerster et al. (2016). We demonstrate that the communication protocols trained with the influence reward meaningfully relate to agents' actions, and that once again, agents trained with the influence reward achieve better collective outcomes.

- We demonstrate that there is a significant correlation between being influenced through communication messages and obtaining higher individual return, suggesting that influential communication is beneficial to the agents that receive it.

- Finally, rather than computing social influence using a centralised training framework as in prior work (e.g. Foerster et al. (2017; 2016)), we extend the approach by attaching an internal *Model of Other Agents* (MOA) network to each agent and training it to predict the actions of every other agent. The agent can then simulate counterfactual actions and use its own internal MOA to predict how these will affect other agents, thus computing its own intrinsic influence reward.

Using a MOA to predict and reward influence allows us to compute an intrinsic social reward by observing other agents' past actions, *without a centralised controller, and without requiring access to another agent's reward function*. We believe this is an important innovation over prior work (e.g. (Hughes et al., 2018; Foerster et al., 2017; 2016)). When we consider likely future applications of MARL, such as autonomous driving, it becomes apparent that centralised training or the sharing of reward functions are unrealistic assumptions, since autonomous vehicles are likely to be produced by a wide variety of organizations and institutions with mixed motivations. Rather, a social reward function which only depends on observing the behavior of agents acting in the environment, and which can give rise to coordinated, cooperative behavior, represents a more promising approach.

## 2 METHODS

We consider a MARL Markov game defined by the tuple $\langle S, T, A, r \rangle$, in which multiple agents which do not share weights are trained to independently maximize their own individual reward. The environment state is given by $s \in S$. At each timestep $t$, each agent $k$ chooses an action $a_t^k \in A$. The actions of all $N$ agents are combined to form a joint action $\boldsymbol{a}_t = [a_t^0, ... a_t^N]$, which produces a transition in the environment $T(s_{t+1}|\boldsymbol{a}_t, s_t)$, according to the state transition function $T$. Each agent then receives its own reward $r^k(\boldsymbol{a}_t, s_t)$, which may depend on the actions of other agents. A history of these variables over time is termed a trajectory, $\tau = \{s_t, \boldsymbol{a}_t, \boldsymbol{r}_t\}_{t=0}^T$. We consider a partially observable setting in which each agent $k$ can only view a portion of the true state, $s_t^k$. Each agent seeks to maximize its own total expected future reward, $R^k = \sum_{i=0}^{\infty} \gamma^i r_{t+i}^k$, where $\gamma$ is a discount factor. A distributed asynchronous advantage actor-critic approach (A3C) (Mnih et al., 2016) is used to train each agent's independent policy $\pi^k$. The policy is learned via REINFORCE with baseline (Williams, 1992). Architecturally, our agents consist of a convolutional layer, fully connected layers, a Long Short Term Memory (LSTM) network (Gers et al., 1999), and linear layers which output $\pi^k$ and the value function $V^{\pi_k}(s)$. We will refer to the internal LSTM state of agent $k$ at timestep $t$ as $u_t^k$.

### 2.1 INTRINSIC SOCIAL MOTIVATION VIA CAUSAL INFLUENCE

Social influence intrinsic motivation modifies an agent's reward function so that it becomes $R^k = \alpha E^k + \beta I^k$, where $E^k$ is the extrinsic or environmental reward, and $I^k$ is the causal influence reward. We compute $I^k$ by generating counterfactual actions that the agent could have taken at each timestep, and assessing how taking these would have affected other agents' behavior. A counterfactual is the estimated probability that "$Y$ would be $y$ had $X$ been $x$, in situation $Z = z$", where $X, Y$, and $Z$ are random variables, and $x, y$ and $z$ are their values (Pearl et al., 2016). Importantly, it is a counterfactual because we condition on a set of evidence $z$, and because the assignment $X = x$ is counter to what we actually observed; in reality, $X$ took on some other value.

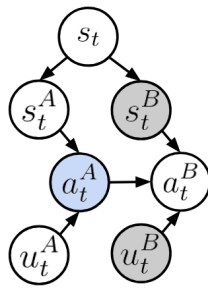

Figure 1: Causal diagram of agent $A$'s effect on $B$'s action. We condition on each agent's view of the environment and LSTM state $u$ (shaded nodes), and intervene on $a_t^A$ (blue).

To see how we can compute the causal effect of one agent on another, suppose there are two agents, $A$ and $B$, and that agent $B$ receives $A$'s action at time $t$, $a_t^A$, as input[1]. Agent $B$ then uses this to compute a distribution over its own action, $p(a_t^B|a_t^A, s_t^B, u_t^B)$. Because we have built the model for agent $B$, we know all of its inputs: $a_t^A$, $s_t^B$, and its own internal LSTM state, $u_t^B$, as shown in Figure 1. This allows us to exactly isolate the causal effect of $A$'s action on $B$ by conditioning on the values we observed for the other inputs at this timestep (note that by conditioning on these variables, including the LSTM state $u_t^B$, we remove any dependency on previous timesteps in the trajectory). We can then intervene on $a_t^A$ by replacing it with a counterfactual action, $do(\tilde{a}_t^A)$. This counterfactual action is used to compute a new estimate of $p(a_t^B|do(\tilde{a}_t^A), s_t^B, u_t^B)$. Essentially, the agent asks a retrospective question: "How would $B$'s action change if I had acted differently in this situation?".

To simplify notation, let $z_t = \langle u_t^B, s_t^B \rangle$, so that conditioning on $z_t$ is equivalent to conditioning on all relevant background variables (all shaded variables in Figure 1). We can also forego the *do* operator, noting that $p(a_t^B|do(\tilde{a}_t^A), z_t) \equiv p(a_t^B|\tilde{a}_t^A, z_t)$ in this case, because $z$ satisfies the back-door criterion (Pearl & Mackenzie, 2018). Now, consider averaging over several counterfactuals $\tilde{a}_t^A$. This gives us the marginal policy of $B$, $p(a_t^B|z_t) = \sum_{\tilde{a}_t^A} p(a_t^B|z_t, \tilde{a}_t^A) p(\tilde{a}_t^A|z_t)$ —in other words, $B$'s policy if $A$ were not considered. The discrepancy between the marginal policy of $B$ and the conditional policy of $B$ given $A$'s action is a measure of the causal influence of $A$ on $B$; it gives the degree to which $B$ changes its planned action distribution because of $A$'s behavior. Thus, the causal influence intrinsic reward for agent $A$ is

$$I_t^A = D_{KL}\Big[ p(a_t^B|a_t^A, z_t) \Big\| \sum_{\tilde{a}_t^A} p(a_t^B|z_t, \tilde{a}_t^A) p(\tilde{a}_t^A|z_t) \Big] = D_{KL}\Big[ p(a_t^B|a_t^A, z_t) \Big\| p(a_t^B|z_t) \Big]. \quad (1)$$

## 2.2 Relationship to Mutual Information and Empowerment

The causal influence reward in Eq. 1 is related to the mutual information (MI) between the actions of agents $A$ and $B$, which is given by

$$I(A^B; A^A|z) = \sum_{a^A, a^B} p(a^B, a^A|z) \log \frac{p(a^B, a^A|z)}{p(a^B|z)p(a^A|z)} = \sum_{a^A} p(a^A|z) D_{\text{KL}}\Big[ p(a^B|a^A, z) \Big\| p(a^B|z) \Big], \quad (2)$$

where we see that the $D_{KL}$ factor in Eq. 2 is the causal influence reward given in Eq. 1. The connection to mutual information is interesting, because a frequently used intrinsic motivation for single agent RL is *empowerment*, which rewards the agent for having high mutual information between its actions and the future state of the environment (e.g. Klyubin et al. (2005); Capdepuy et al. (2007)). To the extent that the social influence reward defined in Eq. 1 is an approximation of the MI, $A$ is rewarded for having empowerment over $B's$ actions.

By sampling $N$ independent trajectories $\tau_n$ from the environment, where A's actions $a_n^A$ are drawn according to $p(a^A|z)$, we perform a Monte-Carlo approximation of the MI (see e.g. Strouse et al. (2018)),

$$I(A^A; A^B|z) = \mathbb{E}_\tau\Big[ D_{\text{KL}}\big[ p(A^B|A^A, z) \big\| p(A^B|z) \big] \Big| z \Big] \approx \frac{1}{N} \sum_n D_{\text{KL}}\big[ p(A^B|a_n^A, z) \big\| p(A^B|z) \big]. \quad (3)$$

Thus, in expectation, the social influence reward is the MI between agents' actions.

Whether the policy trained with Eq. 1 actually learns to approximate the MI depends on the learning dynamics. We calculate the intrinsic social influence reward using Eq. 1, because unlike Eq. 2, which gives an estimate of the symmetric bandwidth between $A$ and $B$, Eq. 1 gives the directed causal effect

---

[1]Note that this requires that agent $A$ choose its action before $B$, and therefore $A$ can influence $B$ but $B$ cannot influence $A$; in other words, we must impose a sequential ordering on agents' actions, and there cannot be mutual influence. We improve upon this approach in Section 2.4. For now, we allow only a fixed number of agents ($\in [1, N-1]$) to be influencers, and the rest are influencees. Only an influencer gets the causal influence reward, and only an influencee can be influenced. At each timestep, the influencers choose their actions first, and these actions are then given as input to the influencees. If agent A and B are influencers, and C is an influencee, then C receives both $a_t^A$ and $a_t^B$ as input. When computing the causal influence of A on C, we also add $a_t^B$ to the conditioning set.

of the specific action taken by agent $A$, $a_t^A$. We believe this will result in an easier reward to learn, since it allows for better credit assignment; agent $A$ can more easily learn which of its actions lead to high influence. We also experiment with replacing the KL-divergence with several other measures, including the Jensen-Shannon Divergence (JSD), and find that the influence reward is robust to the choice of measure.

## 2.3 INFLUENCE THROUGH COMMUNICATION

According to Melis & Semmann (2010), human children rapidly learn to use communication to influence the behavior of others when engaging in cooperative activities. They explain that "this ability to influence the partner via communication has been interpreted as evidence for a capacity to form shared goals with others", and that this capacity may be "what allows humans to engage in a wide range of cooperative activities". Therefore, we investigate a second use of the social influence reward: learning inter-agent communication protocols. Using a similar approach to Reinforced Inter-Agent Learning (RIAL) (Foerster et al., 2016), we equip the agents with an explicit communication channel. At each timestep, each agent $k$ chooses a discrete communication symbol $m_t^k$; these symbols are concatenated into a combined message vector $\boldsymbol{m}_t = [m_t^0, m_t^1...m_t^N]$, for $N$ agents. This message vector $\boldsymbol{m}_t$ is then shown to every other agent in the next timestep, as in Figure 2. To train the agents to communicate, we augment our initial network with an additional A3C output head, that learns a communication policy $\pi_c$ over which symbol to emit, and a communication value function $V_c$ (this is separate from the normal policy and value function used for acting in the environment, $\pi_e$ and $V_e$, which are trained only with environmental reward $E$).

The influence reward is used, in addition to environmental reward, to train the communication policy $\pi_c$. Counterfactuals are employed to assess how much influence an agent's communication message, $m_t^A$, has on another agent's action in the next timestep, $a_{t+1}^B$. Importantly, we hypothesize that communication can only be influential if it is useful to another agent. There is nothing that compels agent $B$ to act based on agent $A$'s communication message; if it does not contain valuable information, $B$ is free to ignore it. In fact, previous work has shown that selfish agents do not learn to use this type of ungrounded, *cheap talk* communication channel effectively (Cao et al., 2018). In contrast, for $A$ to gain influence via communication, $m_t^A$ must contain valuable information that informs $B$ about how best to maximize its own reward, so much so that it actually causes $B$ to change its intended action.

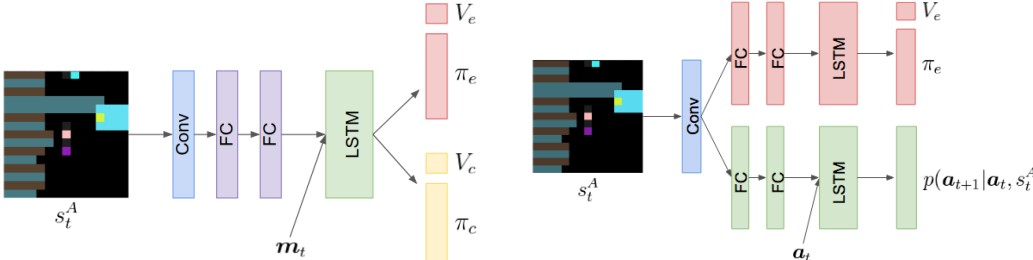

Figure 2: The communication model has two A2C heads, which learn a normal policy, $\pi_e$, and a policy for emitting communication symbols, $\pi_c$. Other agents' communication messages $\boldsymbol{m}_t$ are input to the LSTM.

Figure 3: The Model of Other Agents (MOA) architecture learns both an RL policy $\pi_e$, and a supervised model that predicts the actions of other agents, $\boldsymbol{a}_{t+1}$. The predictions of the supervised model are used for computing the influence reward.

## 2.4 INFLUENCE VIA MODELING OTHER AGENTS

Computing the causal influence reward as introduced in Section 2.1 requires knowing the probability of $B$'s next action given a counterfactual, $p(a_t^B|\tilde{a}_t^A, s_t^B)$, which we previously solved by using a centralised controller that could access other agent's policy networks. While using a centralised training framework is common in MARL (e.g. Foerster et al. (2017; 2016)), it is less realistic than a scenario in which each agent is trained independently. We can relax this assumption and achieve independent training by equipping each agent with its own internal *Model of Other Agents* (MOA). The MOA consists of a second set of fully-connected and LSTM layers connected to the agent's convolutional layer (see Figure 3), and is trained to predict all other agents' next actions given their current action, and the agent's egocentric view of the state: $p(\boldsymbol{a}_{t+1}|\boldsymbol{a}_t, s_t^A)$. The MOA is trained using cross-entropy loss over observed action trajectories.

A trained MOA can be used to compute the social influence reward in the following way. Each agent can "imagine" counterfactual actions that it could have taken at each timestep, and use its internal MOA to

predict the effect on other agents. It can then give itself reward for taking actions that it estimates were the most influential. This has an intuitive appeal, because it resembles how humans reason about their effect on others (Ferguson et al., 2010). We may often find ourselves asking counterfactual questions of the form, "How would she have reacted if I had said or done something else in that situation?", which we can only answer using our internal model of others.

Both the MOA and communication approaches are an important improvement over the original model shown in Figure 1, which computed influence within a given timestep and required that agent $A$ choose its action $a_t^A$ first, and this action be transmitted to agent $B$ as input. This meant that only some agents (those acting first) could be influencers. In contrast, using influence for communication or with a MOA are general approaches that can be implemented in any agent, and allow all agents to mutually influence each other.

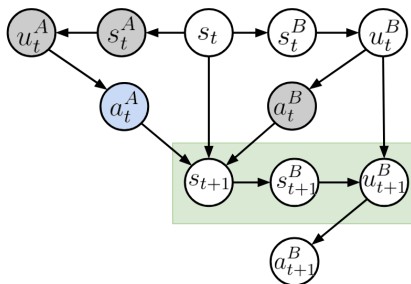

We now seek to estimate influence in the next timestep, meaning the influence of $a_t^A$ on $a_{t+1}^B$, which requires modeling $p(a_{t+1}^B|a_t^A,s_t^A)$. The corresponding causal diagram is shown in Figure 4. We can infer the causal effect of $a_t^A$ on $a_{t+1}^B$ by conditioning on the shaded variables (so that there are no back-door paths) (Pearl et al., 2016). Learning a model of $p(a_{t+1}^B|a_t^A,s_t^A)$ requires implicitly modeling both the environment transition function $T$ (to predict $s_{t+1}$), as well as relevant aspects of the internal LSTM state of the other agent, $u_{t+1}^B$, as highlighted in Figure 4.

Figure 4: Causal diagram in the MOA case. Shaded nodes are conditioned on, and we intervene on $a_t^A$ (blue node) by replacing it with counterfactuals. Nodes with a green background must be modeled using the MOA module. Note that there is no backdoor path between $a_t^A$ and $s_t$ since it would require traversing a collider that is not in the conditioning set.

We enable agents to condition their policy on the actions of other agents in the previous timestep (actions are visible), and only give the social influence reward to an agent when the agent it is attempting to influence is within its field-of-view, because the estimates of $p(a_{t+1}^B|a_t^A,s_t^A)$ are likely to be more accurate when $B$ is visible to $A$[2]. The latter constraint could have the side-effect of encouraging agents to stay in closer proximity. However, an intrinsic social reward based on proximity is also a reasonable approach to approximating human social motivation. Humans seek affiliation and to spend time near other people (Tomasello, 2009).

## 2.5 SEQUENTIAL SOCIAL DILEMMAS

First proposed by Leibo et al. (2017), Sequential Social Dilemmas (SSDs) are spatially and temporally extended multi-agent games that have a payoff structure similar to that of Prisoner's Dilemma (PD). That is, an individual agent can obtain higher reward by engaging in defecting, non-cooperative behavior (and thus is rationally motivated to defect), but the average payoff per agent will be higher if all agents cooperate (see Figure 9 of the Appendix). The paradoxical reward structure makes it extremely difficult for traditional RL agents to learn to coordinate to solve the tasks (Hughes et al., 2018). We experiment with two SSDs in this work, a public goods game *Cleanup*, and a tragedy-of-the-commons game *Harvest* (see Figure 5). In both games apples (green tiles) provide the rewards, and agents also have the ability to punish each other with a *fining beam*. Further details are available in Appendix Section 6.1.

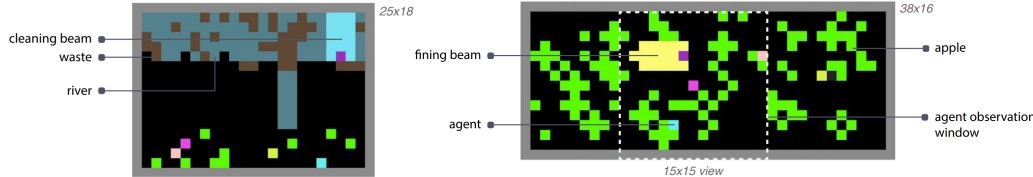

Figure 5: The two SSD environments, *Cleanup* (left) and *Harvest* (right). Agents can exploit other agents for immediate payoff, but at the expense of the long-term collective reward of the group.

---

[2]This contrasts with our previous models in which the influence reward was obtained even from non-visible agents.

## 3    RELATED WORK

Several attempts have been made to develop intrinsic social motivation rewards[3]. Sequeira et al. (2011) developed hand-crafted rewards specific to a foraging environment, in which agents were punished for eating more than their fair share of food. Another approach gave agents an emotional intrinsic reward based on their perception of their neighbours' cooperativeness in a networked version of the iterated prisoner's dilemma (Yu et al., 2013). This approach is limited to scenarios in which it is possible to directly classify each action as cooperative or non-cooperative, which is untenable in complex settings with long-term strategies, such as the SSDs under investigation here. Hughes et al. (2018) introduced an inequity aversion motivation, which penalized agents if their rewards differed too much from those of the group. Another approach used prosocial reward shaping to show that if even a single agent is trained to optimize for the rewards of other agents, it can help the group obtain better collective outcomes (Peysakhovich & Lerer, 2018). However, these both require the ability to observe other agent's rewards, which may be an unrealistic assumption, depending on the application.

Another body of work has focused on training agents to learn emergent communication protocols (Foerster et al., 2016; Cao et al., 2018; Choi et al., 2018; Lazaridou et al., 2018; Bogin et al., 2018), with many authors finding that selfish agents do not learn to use an ungrounded, *cheap talk* communication channel effectively. Crawford & Sobel (1982) find that in theory, the information revealed in communication (in equilibrium) is proportional to amount of common interest; thus, as agents' interests diverge, no communication is to be expected. And while communication can emerge when agents are prosocial (Foerster et al., 2016; Lazaridou et al., 2018) or hand-crafted (Crandall et al., 2017), self-interested agents do not to learn to communicate (Cao et al., 2018). We test whether the social influence reward can encourage agents to learn to communicate more effectively in complex environments with challenging social dilemma dynamics.

Interestingly, Oudeyer & Kaplan (2006) show that a robot trained with a curiosity-based intrinsic motivation to maximize learning progress learns to prefer vocalizing sounds imitated by another robot over interaction with other objects in the environment. Follow-up papers suggest that curiosity may be a sufficient motivation to encourage agents, or even children, to learn to communicate with others (Oudeyer & Smith, 2016; Forestier & Oudeyer, 2017).

Our MOA network is related to work on machine theory of mind (Rabinowitz et al., 2018), which demonstrated that a model trained to predict agents' actions is able to model false beliefs. With LOLA, Foerster et al. (2018) train agents that model the impact of their policy on the parameter updates of other agents, and directly incorporate this into the agent's own learning rule.

Barton et al. (2018) propose causal influence as a way to measure coordination between agents, specifically using Convergence Cross Mapping (CCM) to analyze the degree of dependence between two agents' policies. The limitation of this approach is that CCM estimates of causality are known to degrade in the presence of stochastic effects (Tajima et al., 2015). Counterfactual reasoning has also been used in a multi-agent setting, to marginalize out the effect of one agent on a predicted global value function estimating collective reward, and thus obtain an improved baseline for computing each agent's advantage function (Foerster et al., 2017). A similar paper shows that counterfactuals can be used with potential-based reward shaping to improve credit assignment for training a joint policy in multi-agent RL Devlin et al. (2014). However, once again these approaches rely on a centralised controller.

Following in the tradition of the empowerment literature, authors have investigated mutual information (MI) as a powerful tool for designing social rewards. Strouse et al. (2018) train agents to maximize or minimize the MI between their actions and a categorical goal, and show how this can be used to signal or hide the agent's intentions. However, this approach depends on agents pursuing a known, categorical goal. Guckelsberger et al. (2018), in pursuit of the ultimate video game adversary, develop an agent that maximizes its empowerment over its own states, minimizes the player's empowerment over their states, and maximizes its empowerment over the player's next state. This third goal, termed *transfer empowerment*, is obtained by maximizing the MI between the agent's actions and the player's future state. While similar to our approach, the authors find that agents trained with transfer empowerment simply tend to stay near the player. Further, the agents are not trained with RL, but rather analytically compute these measures in simple grid-world environments. As such, the agent cannot learn to model other agents or the environment.

---

[3]Note that *intrinsic* is not a synonym of *internal*; it is possible to be intrinsically motivated by other people (Stavropoulos & Carver, 2013).

## 4 EXPERIMENTS

The following sections present the results of training agents with the social influence reward in three settings: (1) using a centralised controller, (2) using an explicit communication channel, and (3) using a learned model of other agents (MOA). In each case we compare against a standard A3C agent, and an ablated version of the model which is architecturally identical, but does not receive the influence reward. We measure the total collective reward obtained using the best hyperparameter setting tested with 5 random seeds. It is worth noting that we use a curriculum learning approach which gradually increases the weight of the social influence reward over $C$ steps ($C \in [0.2-3.5] \times 10^8$); this can lead to a slight delay before the influence models' performance improves.

We also provide the results of an additional experiment Section 6.2 of the Appendix, which tests the social influence reward in a simplified environment where the effects of influence are clear. We encourage the reader to examine that section to gain a better intuition for how social influence can foster cooperative behavior in an otherwise selfish agent.

### 4.1 CENTRALISED CONTROLLER

Figures 6(a) and 6(d) show the results of training influence with a centralised controller as described in Section 2.1. With this method, the influencer agents transmit their intended action to the influenced agents at each timestep. Therefore, we benchmark against an ablated version of the influence model with visible actions but no influence reward. As is evident in Figures 6(a) and 6(d), introducing an awareness of other agents' actions helps, but having the social influence reward eventually leads to significantly higher collective reward in both games.

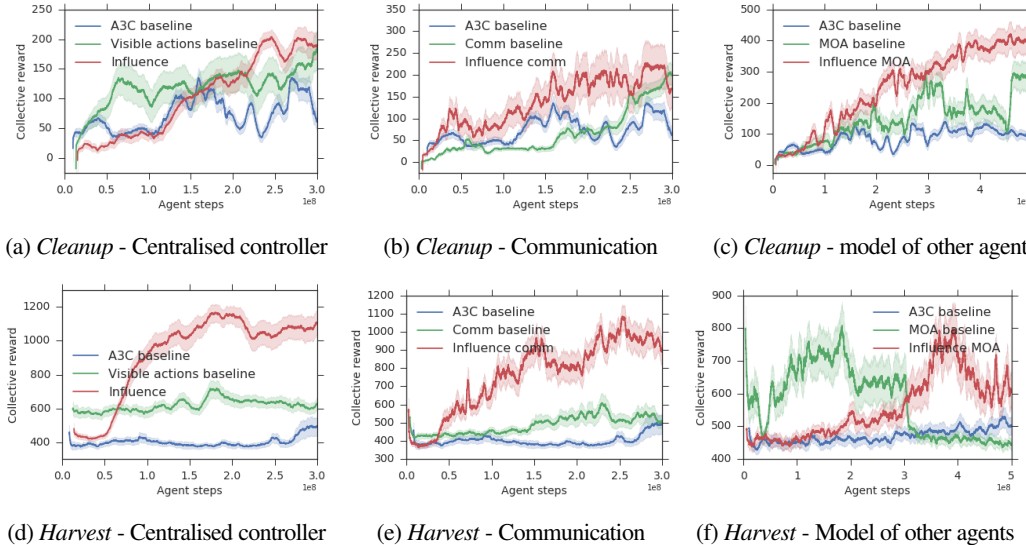

(a) *Cleanup* - Centralised controller  (b) *Cleanup* - Communication  (c) *Cleanup* - model of other agents

(d) *Harvest* - Centralised controller  (e) *Harvest* - Communication  (f) *Harvest* - Model of other agents

Figure 6: Total collective reward obtained in all experiments. Error bars show a 99.5% confidence interval (CI) over 5 random seeds, computed within a sliding window of 200 agent steps. The models trained with influence reward (red) significantly outperform the baseline and ablated models.

While these aggregated results demonstrate the success of our models, they are not sufficient to understand the mechanism through which social influence is helping the agents achieve cooperative behavior. Therefore, we investigated the trajectories produced by high scoring models in both *Cleanup* and *Harvest*; the analysis revealed interesting behavior. As an example, in the *Cleanup* video available here: `https://youtu.be/iH_V5WKQxmo` a single agent (shown in purple) was trained with the social influence reward. We see that unlike the other agents, which continue to randomly move and explore while waiting for apples to spawn, the influencer has a strange economy of motion; it only moves on the map when it is pursuing an apple, then stops. Interestingly, examining the trajectory reveals that the influencer uses only two moves to explore the map: *turn left*, which turns the agent in place without traversing the map, and *move right*, which moves the agent one square to the right on the map.

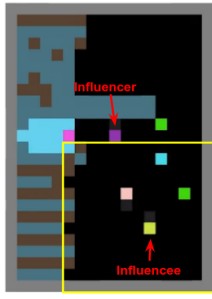

Figure 7: A moment of high influence when the purple influencer signals the presence of an apple outside the yellow influencee's field-of-view (yellow outlined box).

Why did the influencer learn to use only these two moves? We can see that the influencer agent only chooses to move right (*i.e.* traverse the map) when it is pursuing an apple which is present. The rest of the time it simply turns left on the spot. At $t = 49$, there is a moment of high influence between the influencer and the yellow influencee, which is shown in Figure 7. The influencer has chosen to *move right* towards an apple that is outside of the ego-centric field-of-view of the yellow agent. Because the purple agent only moves when apples are available, this signals to the yellow agent that an apple must be present above it which it cannot see. This changes the yellow agent's distribution over its planned action, $p(a_t^B | a_t^A, s_t^B)$, and allows the purple agent to gain influence. A similar moment occurs when the influencer signals to an agent that has been cleaning the river that no apples have appeared by continuing to *turn left* (see Figure 12 in the Appendix).

In this example, the influencer agent learned to use its own actions as a sort of binary code, which signals the presence or absence of apples in the environment. We also observe this effect in the influence agents in the *Harvest* task. This type of action-based communication could be likened to the bee waggle dance discovered by von Frisch (1969). Thus, rewarding agents for increasing the mutual information between their actions gave rise not only to cooperative behavior, but in this case, to emergent communication. These results further support the idea of using influence as a reward for training agents to communicate.

## 4.2 INFLUENCE THROUGH COMMUNICATION

Figures 6(b) and 6(e) show the results of training the agents to use an explicit communication channel, and its effect on their collective reward. In this case, the ablated baseline is a model that has the same structure as in Figure 2, but in which the communication policy $\pi_c$ is trained only with environmental reward. We observe that the agents which are trained to use the communication channel with additional social influence reward achieve significantly higher collective reward in both games. In fact, in the case of *Cleanup*, we found that $\alpha = 0$ in the optimal hyperparameter settings, meaning that it was most effective to train the communication head with zero extrinsic or environmental reward (see Table 2 in the Appendix). This suggests that influence alone can be a sufficient mechanism for training an effective communication policy.

To analyze the communication behaviour learned by the agents, we introduce three metrics. *Speaker consistency*, is a normalized score $\in [0, 1]$ which assesses the entropy of $p(a^k | m^k)$ and $p(m^k | a^k)$ to determine how consistently a *speaker* agent emits a particular symbol when it takes a particular action, and vice versa (the formula is given in Appendix Section 6.3.4). We expect this measure to be high if, for example, the speaker always emits the same symbol when it is cleaning the river. We also introduce two measures of *instantaneous coordination* (IC), which are both measures of mutual information (MI): (1) symbol/action IC $= I(m_t^A; a_{t+1}^B)$ measures the MI between the influencer/speaker's symbol and the influencee/listener's next action, and (2) action/action IC $= I(a_t^A; a_{t+1}^B)$ measures the MI between the influencer's action and the influencee's action in the next timestep. To compute these measures we first average over all trajectory steps, then take the maximum value between any two agents, to determine if any pair of agents are coordinating. Note that these measures are all *instantaneous*, as they consider only short-term dependencies across two consecutive timesteps, and cannot capture if an agent communicates influential compositional messages, i.e. information that requires several consecutive symbols to transmit and only then affects the other agents behavior.

Figure 8 presents the results. The speaker consistencies metric reveals that agents trained with the influence reward communicate less ambiguously about their own actions, indicating that the emergent communication is more meaningful. The instantaneous coordination metrics demonstrate

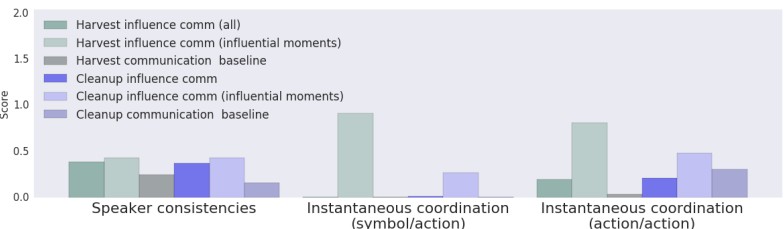

Figure 8: Metrics describing the quality of learned communication protocols. The models trained with influence reward exhibit more consistent communication and more coordination, especially in moments where influence is high.

that the baseline agents trained without influence reward show almost no signs of co-ordinating behavior with communication, i.e. speakers saying A and listeners doing B consistently. This result is aligned with both theoretical results in cheap-talk literature (Crawford & Sobel, 1982), and recent empirical results in MARL (e.g. (Foerster et al., 2016; Lazaridou et al., 2018; Cao et al., 2018)).

In contrast, we do see highly coordinated behavior between influence agents, but only when we limit the analysis to timesteps on which influence was high (cf. *influential moments* in Figure 8). If we inspect the results for agents trained with influence on the two tasks, a common pattern emerges: influence is sparse in time. An agent's influence is only greater than its mean influence in less than 10% of timesteps. Because the listener agent is not compelled to listen to any given speaker, listeners selectively listen to a speaker only when it is beneficial, and influence cannot occur all the time. Only when the listener decides to change its action based on the speaker's message does influence occur, and in these moments we observe high $I(m_t^A; a_{t+1}^B)$; an effect that is lost when averaging over the entire trajectory. It appears the influencers have learned a strategy of communicating meaningful information about their own actions, and gaining influence when this becomes relevant enough for the listener to act upon it.

Examining the relationship between the reward obtained by individual agents and the degree to which they were influenced by other agents gives a compelling result: agents that are the most influenced also achieve higher individual environmental reward, $E^k$. We sampled 100 different experimental conditions (i.e., hyper-parameters and random seeds) for both games, collected the influence and individual rewards, normalized them across the 5 agents in each condition, and correlated the resulting list of values. We found that agents who are more often influenced tend to achieve higher task reward in both *Cleanup*, $\rho = .67$, $p<0.001$, and *Harvest*, $\rho = .34$, $p<0.001$. This supports the hypothesis stated in Section 2.3: in order to gain influence from another agent by communicating with it, the communication message should contain information that helps the listener maximize its own environmental reward. Since better listeners/influencees are more successful in terms of task reward, we have evidence that useful information was transmitted to them.

### 4.3 INFLUENCE VIA MODELING OTHER AGENTS

Finally, we investigate whether the influence reward is still effective when computed without a centralised controller, but rather through each agent's own internal *Model of Other Agents* (MOA) network. In this case, we extend the training period from $3 \cdot 10^8$ steps to $5 \cdot 10^8$, in order to give the MOA model time to train. We also allow the policy LSTM to condition on the actions of other agents in the last timestep. We compare against an ablated version of this architecture (shown in Figure 3), which does not use the output of the MOA module to compute a reward; rather, the MOA module can be thought of as an unsupervised auxiliary task that may help the model to learn a better shared embedding layer, encouraging it to encode information relevant to predicting other agents' behavior.

Figures 6(c) and 6(f) show the collective reward obtained for agents trained with a MOA module. While we see that the auxiliary task does help to improve reward over the A3C baseline, the influence agent gets consistently higher collective reward. Impressively, for *Cleanup*, the MOA model scores higher than the original influence agents computed using the centralised controller (CC). As shown in Figure 6(c), the MOA baseline also achieves high collective reward, suggesting that the auxiliary task of modeling other agents helps the MOA agents cooperate more effectively in *Cleanup*. Further, the independent design of the MOA method allows each agent to influence every other agent, thus generating more reward signal and a greater chance to develop two-way cooperative behavior.

Table 4 of the Appendix gives the final collective reward obtained by each model for all three experiments. Interestingly, several influence models are able to achieve higher collective reward than the previous state-of-the-art scores for these environments (275 for *Cleanup* and 750 for *Harvest*) (Hughes et al., 2018). This is compelling, given that previous work relied on the assumption that agents could view one another's rewards; we make no such assumption, instead relying only on agents viewing each other's actions.

## 5 DISCUSSION AND CONCLUSIONS

The experiments above have demonstrated that an intrinsic social reward based on having causal influence on the actions of other agents consistently improves cooperation and leads to higher collective return in the MA social dilemmas under investigation. In some cases, the influence reward drove agents to learn an emergent communication protocol via their actions. This is compelling, and confirms the connection between maximizing influence and maximizing the mutual information between agents' actions.

However, it is important to consider the limitations of the influence reward. Whether it will always give rise to cooperative behavior may depend on the specifics of the environment, task, and the trade-off between environmental and influence reward. Although influence is arguably necessary for cooperation (e.g. two agents cooperating to lift a box would have a high degree of influence between their actions), it may not be sufficient, in that it may be possible to influence another agent without helping it. For example, it is possible that agents could have gained influence in the tasks studied here by threatening to attack other agents with their fining beam. We believe this type of behavior did not emerge because communicating information represents the cheapest and most effective way to gain influence. Influencers do not have to sacrifice much in terms of their own environmental reward in order to communicate to other agents.

Rewarding influence over an explicit communication channel may not be subject to this limitation, because influential communication may be inherently beneficial to the listener (at least in the case where listeners and speakers interact repeatedly). Since listeners can easily ignore communication messages if they do not help to obtain environmental reward, a speaker must transmit valuable information in order to gain influence through communication. There is no advantage to the speaker for communicating unreliably, because it would lose influence with the listener over time (although this is no longer guaranteed in one-shot interactions). Indeed, our results reveal that agents benefit from being influenced by (listening to) communication messages by obtaining higher individual reward, suggesting that the messages contain valuable information. Further, we found that the communication protocols learned via influence reward were more meaningful, and that the influence reward allowed agents to obtain higher collective return. Therefore, we suggest that influence could be a promising way to train emergent communication protocols in various settings.

Finally, we have shown that influence can be computed by augmenting agents with an internal model that predicts the actions of other agents, and using this MOA model to simulate the effect of an agent's actions on others. This represents an important step forward in multi-agent intrinsic social motivation, because it implies that the influence reward can be computed without having access to another agent's reward function, or requiring a centralised controller.

## 5.1 FUTURE WORK

Using counterfactuals to allow agents to understand the effects of their actions on other agents could be a promising approach with a number of extensions. Perhaps agents could use counterfactuals to develop a form of 'empathy', by simulating how their actions affect another agent's value function. Or, social influence could be used to drive coordinated behavior in robots attempting to do cooperative manipulation and control tasks. Finally, if we view multi-agent networks as a single agent, influence could be used as a regularizer to encourage different modules of the network to integrate information from other networks; for example, perhaps it could prevent collapse in hierarchical RL.

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

## 6 APPENDIX

### 6.1 SEQUENTIAL SOCIAL DILEMMAS

In each of the sequential social dilemma (SSD) games studied above, an agent is rewarded +1 for every apple it collects, but the apples are a limited resource. In *Harvest* (a tragedy of the commons game), apples regenerate more slowly the faster they are harvested, and if an exploiting agent consumes all of the apples, they will not grow back; agents must cooperate to harvest sustainably. In *Cleanup* (a public goods game), apples are generated based on the amount of waste in a nearby river. Agents can use a *cleaning beam* action to clean the river when they are positioned in it; or they can simply consume the apples the other agent produces. Agents also have a *fining beam* action which they can use to fine nearby agents −50 reward.

Figure 9 gives the Schelling diagram for both SSD tasks under investigation. A Schelling diagram (Schelling, 1973; Perolat et al., 2017) shows the relative payoffs for a single agent's strategy given a fixed number of other agents who are cooperative. Schelling diagrams generalize payoff matrices to multi-agent settings, and make it easy to visually recognize game-theoretic properties like Nash equilibria (see Schelling (1973) for more details).

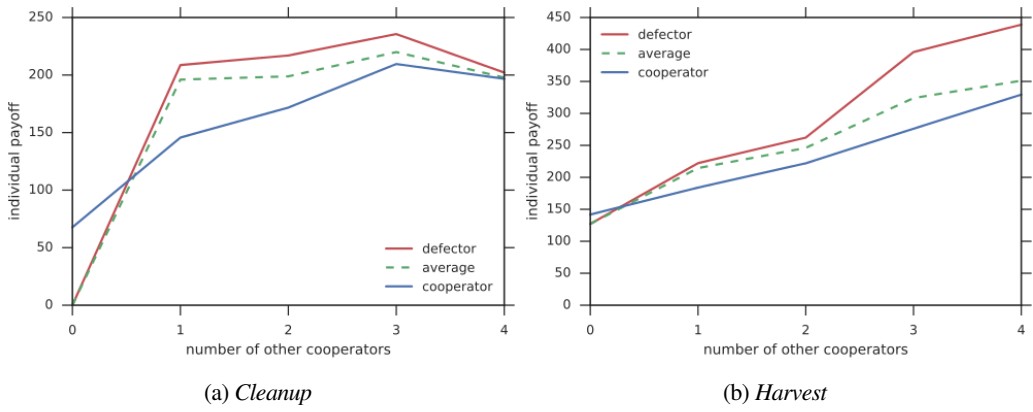

(a) *Cleanup*                                        (b) *Harvest*

Figure 9: Schelling diagrams for the two social dilemma tasks show that an individual is almost always motivated to defect, even though the group will get higher reward if there are more cooperators.

### 6.2 ADDITIONAL EXPERIMENT - BOX TRAPPED

As a proof-of-concept experiment to test whether the influence reward works as expected, we constructed a special environment, shown in Figure 10. In this environment, one agent (teal) is trapped in a box. The other agent (purple) has a special action it can use to open the box... or it can simply choose to consume apples, which exist outside the box and are inexhaustible in this environment.

As expected, a vanilla A3C agent learns to act selfishly; the purple agent will simply consume apples, and chooses the *open box* action in 0% of trajectories once the policy has converged. A video of A3C agents trained in this environment is available at: https://youtu.be/C8SE9_ YKzxI, which shows that the purple agent leaves its compatriot trapped in the box throughout the trajectory.

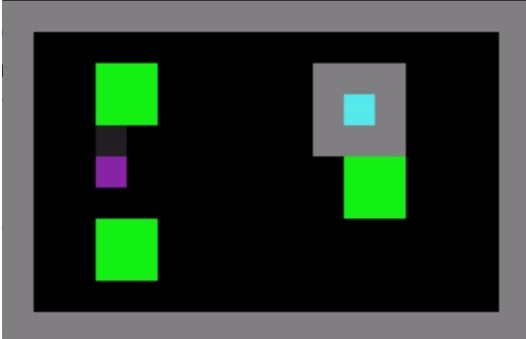

Figure 10: The *Box trapped* environment in which the teal agent is trapped, and the purple agent can release it with a special *open box* action.

In contrast, an agent trained with the social influence reward chooses the *open box* action in 88% of trajectories, releasing its fellow agent so that they are both able to consume apples. A video of this behavior is shown at: https://youtu.be/Gfo248-qt3c. Further, as Figure 11(a) reveals, the purple influencer agent usually chooses to open the box within the first few steps of the trajectory, giving its fellow agent more time to collect reward.

Most importantly though, Figure 11(b) shows the influence reward over the course of a trajectory in the *Box trapped* environment. The agent chooses the *open box* action in the second timestep; at this point, we see a corresponding spike in the influence reward. This reveals that the influence reward works as expected, incentivizing an action which has a strong — and in this case, prosocial — effect on the other agent's behavior.

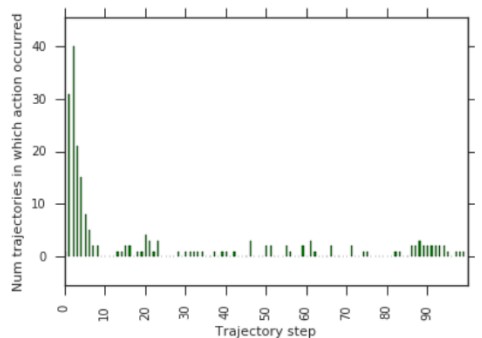

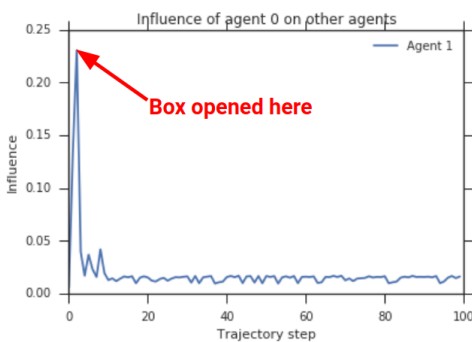

(a) Number of times the *open box* action occurs at each trajectory step over 100 trajectories.

(b) Influence reward over a trajectory in *Box trapped*

Figure 11: The *Box trapped* proof-of-concept experiment reveals that an agent gets high influence for letting another agent out of a box in which it is trapped.

### 6.3 IMPLEMENTATION DETAILS

All models are trained with a single convolutional layer with a kernel of size 3, stride of size 1, and 6 output channels. This is connected to two fully connected layers of size 32 each, and an LSTM with 128 cells. We use a discount factor $\gamma = .99$. The number of agents $N$ is fixed to 5.

As mentioned in Section 2.2, the social influence reward can be computed using a number of divergence measures, including JSD. We also experiment with training the agents using the *pointwise mutual information* (the innermost term of Eq. 3), which is given by:

$$pmi(a^A; a^B \mid Z = z) = \log \frac{p(a^B \mid a^A, z)}{p(a^B \mid z)} = \log \frac{p(a^A, a^B \mid z)}{p(a^A \mid z)p(a^B \mid z)}. \tag{4}$$

This PMI term is precisely the *local information flow* proposed by Lizier & Prokopenko (2010) as a measure of direct causal effect; the expectation of the PMI over $p(a^B, a^A \mid z)$ is the MI. and gives us a measure of influence of a single action of $A$ on the single action taken by $B$.

In addition to the comparison function used to compute influence, there are many other hyperparameters that can be tuned for each model. We use a random search over hyperparameters, ensuring a fair comparison with the search size over the baseline parameters that are shared with the influence models. For all models we search for the optimal entropy reward and learning rate, where we anneal the learning rate from an initial value `lr_init` to `lr_final`. The below sections give the parameters found to be most effective for each of the three experiments.

### 6.3.1 CENTRALISED CONTROLLER HYPERPARAMETERS

In this setting we vary the number of influencers from $1-4$, the influence reward weight $\beta$, and the number of curriculum steps over which the weight of the influence reward is linearly increased $C$. In this setting, since we have a centralised controller, we also experiment with giving the influence reward to the agent being influenced as well, and find that this sometimes helps. This 'influencee' reward is not used in the other two experiments, since it precludes independent training. The hyperparameters found to give the best performance for each model are shown in Table 1.

### 6.3.2 COMMUNICATION HYPERPARAMETERS

Because the communication models have an extra A2C output head for the communication policy, we use an additional entropy regularization term just for this head, and apply a weight to the communication loss

| Hyperparameter | Cleanup | | | Harvest | | |
|---|---|---|---|---|---|---|
| | A3C baseline | Visible actions baseline | Influence | A3C baseline | Visible actions baseline | Influence |
| Entropy reg. | .00176 | .00176 | .000248 | .000687 | .00184 | .00025 |
| lr_init | .00126 | .00126 | .00107 | .00136 | .00215 | .00107 |
| lr_end | .000012 | .000012 | .000042 | .000028 | .000013 | .000042 |
| Number of influencers | - | 3 | 1 | - | 3 | 3 |
| Influence weight $\beta$ | - | 0 | .146 | - | 0 | .224 |
| Curriculum $C$ | - | - | 140 | - | - | 140 |
| Policy comparison | - | - | JSD | - | - | PMI |
| Influencee reward | - | - | 1 | - | - | 0 |

Table 1: Optimal hyperparameter settings for the models in the centralised controller experiment.

| Hyperparameter | Cleanup | | | Harvest | | |
|---|---|---|---|---|---|---|
| | A3C baseline | Comm. baseline | Influence comm. | A3C baseline | Comm. baseline | Influence comm. |
| Entropy reg. | .00176 | .000249 | .00305 | .000687 | .000174 | .00220 |
| lr_init | .00126 | .00223 | .00249 | .00136 | .00137 | .000413 |
| lr_end | .000012 | .000022 | .0000127 | .000028 | .0000127 | .000049 |
| Influence weight $\beta$ | - | 0 | 2.752 | - | 0 | 4.825 |
| Extrinsic reward weight $\alpha$ | - | - | 0 | - | - | 1.0 |
| Curriculum $C$ | - | - | 1 | - | - | 8 |
| Policy comparison | - | - | KL | - | - | KL |
| Comm. entropy reg. | - | - | .000789 | - | - | .00208 |
| Comm. loss weight | - | - | .0758 | - | - | .0709 |
| Symbol vocab size | - | - | 9 | - | - | 7 |
| Comm. embedding | - | - | 32 | - | - | 16 |

Table 2: Optimal hyperparameter settings for the models in the communication experiment.

| Hyperparameter | Cleanup | | | Harvest | | |
|---|---|---|---|---|---|---|
| | A3C baseline | MOA baseline | Influence MOA | A3C baseline | MOA baseline | Influence MOA |
| Entropy reg. | .00176 | .00176 | .00176 | .000687 | .00495 | .00223 |
| lr_init | .00126 | .00123 | .00123 | .00136 | .00206 | .00120 |
| lr_end | .000012 | .000012 | .000012 | .000028 | .000022 | .000044 |
| Influence weight $\beta$ | - | 0 | .620 | - | 0 | 2.521 |
| MOA loss weight | - | 1.312 | 15.007 | - | 1.711 | 10.911 |
| Curriculum $C$ | - | - | 40 | - | - | 226 |
| Policy comparison | - | - | KL | - | - | KL |
| Train MOA only when visible | - | False | True | - | False | True |

Table 3: Optimal hyperparameter settings for the models in the model of other agents (MOA) experiment.

in the loss function. We also vary the number of communication symbols that the agents can emit, and the size of the linear layer that connects the LSTM to the communication policy layer, which we term the communication embedding size. Finally, in the communication regime, we experiment to setting the weight on the extrinsic reward E, $\alpha$, to zero. The best hyperparameters for each of the communication models are shown in Table 2.

### 6.3.3 MODEL OF OTHER AGENTS (MOA) HYPERPARAMETERS

The MOA hyperparameters include whether to only train the MOA with cross-entropy loss on the actions of agents that are visible, and how much to weight the supervised loss in the overall loss of the model. The best hyperparameters are shown in Table 3.

|  | Cleanup | Harvest |
|---|---|---|
| A3C baseline | 89 | 485 |
| Inequity aversion (Hughes et al., 2018) | 275 | 750 |
| Influence - Basic | 190 | **1073** |
| Influence - Communication | 166 | **951** |
| Influence - Model of other agents | **392** | 588 |

Table 4: Final collective reward over the last 50 agent steps for each of the models considered. Bolded entries represent experiments in which the influence models significantly outperformed the scores reported in previous work on *inequity aversion*(Hughes et al., 2018). This is impressive, considering the *inequity averse* agents are able to view all other agents' rewards. We make no such assumption, and yet are able to achieve similar or superior performance.

### 6.3.4 COMMUNICATION ANALYSIS

The speaker consistency metric is calculated as:

$$\sum_{k=1}^{N} 0.5[\sum_{c} 1 - \frac{H(p(a^k|m^k=c))}{H_{max}} + \sum_{a} 1 - \frac{H(p(m^k|a^k=a))}{H_{max}}], \tag{5}$$

where $H$ is the entropy function and $H_{max}$ is the maximum entropy based on the number of discrete symbols or actions. The goal of the metric is to measure how much of a 1:1 correspondence exists between a speaker's action and the speaker's communication message.

### 6.4 ADDITIONAL RESULTS

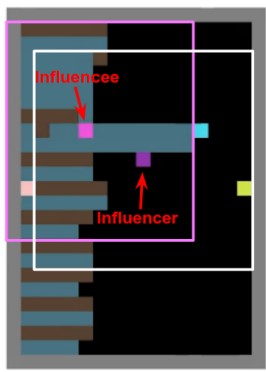

Figure 12: A moment of high influence between the purple influencer and magenta influencee.

Figure 12 shows an additional moment of high influence in the *Cleanup* game. The purple influencer agent can see the area within the white box, and therefore all of the apple patch. The field-of-view of the magenta influencee is outlined with the magenta box; it cannot see if apples have appeared, even though it has been cleaning the river, which is the action required to cause apples to appear. When the purple influencer turns left and does not move towards the apple patch, this signals to the magenta agent that no apples have appeared, since otherwise the influence would move right.

Table 4 presents the final collective reward obtained by each of the models tested in the experiments presented in Section 4. We see that in several cases, the influence agents are even able to out-perform the state-of-the-art results on these tasks reported by Hughes et al. (2018), despite the fact that the solution proposed by Hughes et al. (2018) requires that agents can view other agents' rewards, whereas we do not make this assumption, and instead only require that agents can view each others' actions.

It is important to note that collective reward is not always the perfect metric of cooperative behavior, a finding that was also discovered by Barton et al. (2018) and emphasized by Leibo et al. (2017). In the case, we find that there is a spurious solution to the *Harvest* game, in which one agent fails to learn and fails to collect any apples. This leads to very high collective reward, since it means there is one fewer agent that can exploit the others, and makes sustainable harvesting easier to achieve. Therefore, for the results shown in the paper, we eliminate any random seed in *Harvest* for which one of the agents has failed to learn to collect apples, as in previous work (Hughes et al., 2018).

However, here we also present an alternative strategy for assessing the overall collective outcomes: weighting the total collective reward by an index of equality of the individual returns. Specifically, we compute the Gini coefficient over the $N$ agents' individual returns:

$$G = \frac{\sum_{i=1}^{N}\sum_{j=1}^{N}|r^i - r^j|}{2N\sum_{i=1}^{N} r^i}, \tag{6}$$

which gives us a measure of the inequality of the returns, where $G \in [0,1]$, with $G=0$ indicating perfect equality. Thus, $1-G$ is a measure of equality; we use this to weight the collective reward for each

experiment, and plot the results in Figure 13. Once again, we see that the influence models give the highest final performance, even with this new metric.

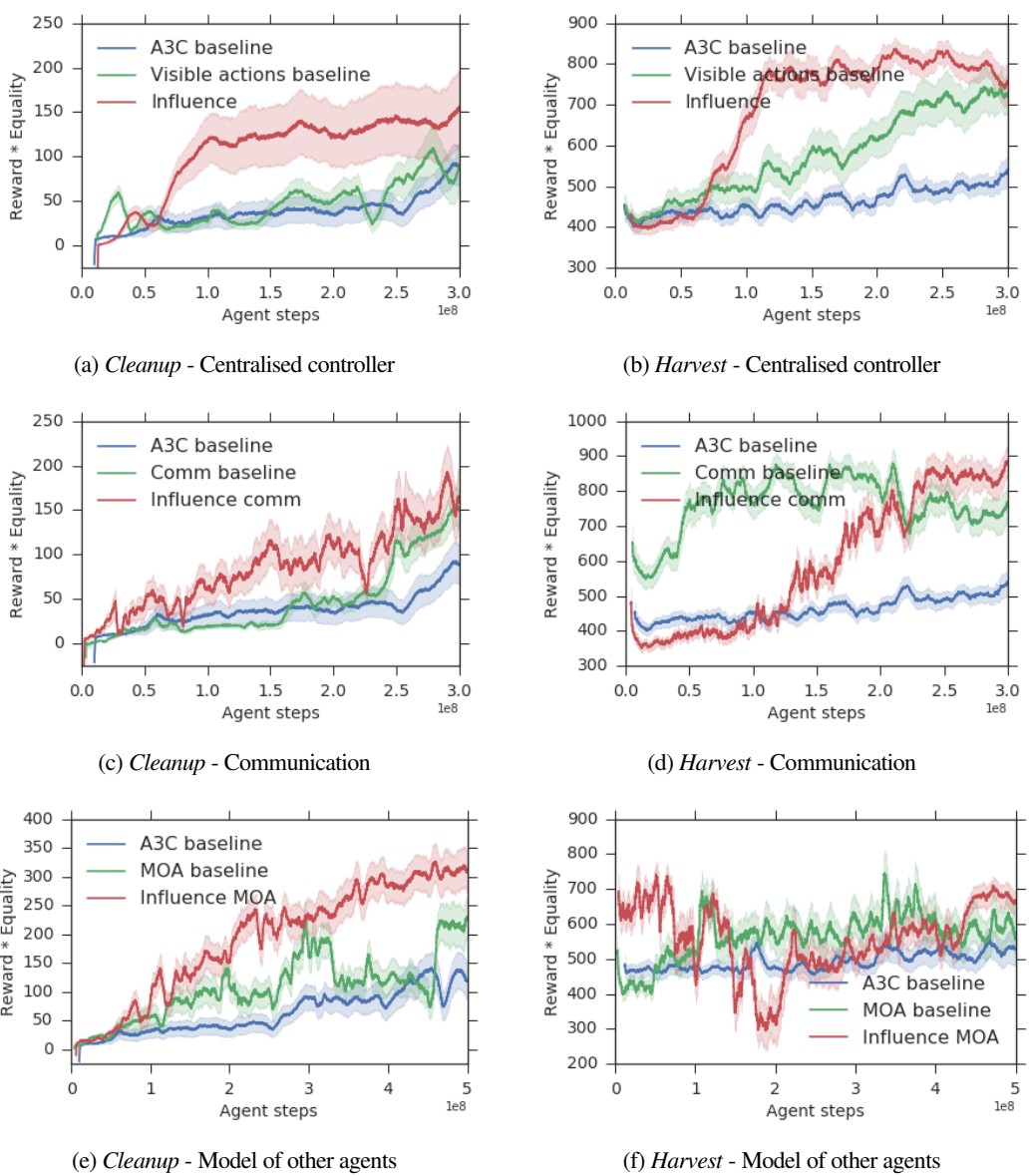

(a) *Cleanup* - Centralised controller

(b) *Harvest* - Centralised controller

(c) *Cleanup* - Communication

(d) *Harvest* - Communication

(e) *Cleanup* - Model of other agents

(f) *Harvest* - Model of other agents

Figure 13: Total collective reward times equality, $R*(1-G)$, obtained in all experiments. Error bars show a 99.5% confidence interval (CI) over 5 random seeds, computed within a sliding window of 200 agent steps. Once again, the models trained with influence reward (red) significantly outperform the baseline and ablated models.

Finally, we would like to show that the influence reward is robust to the choice of hyperparameter settings. Therefore, in Figure 14, we plot the collective reward of the top 5 best hyperparameter settings for each experiment, over 5 random seeds each. Once again, the influence models result in higher collective reward, which provides evidence that the model is robust to the choice of hyperparameters.

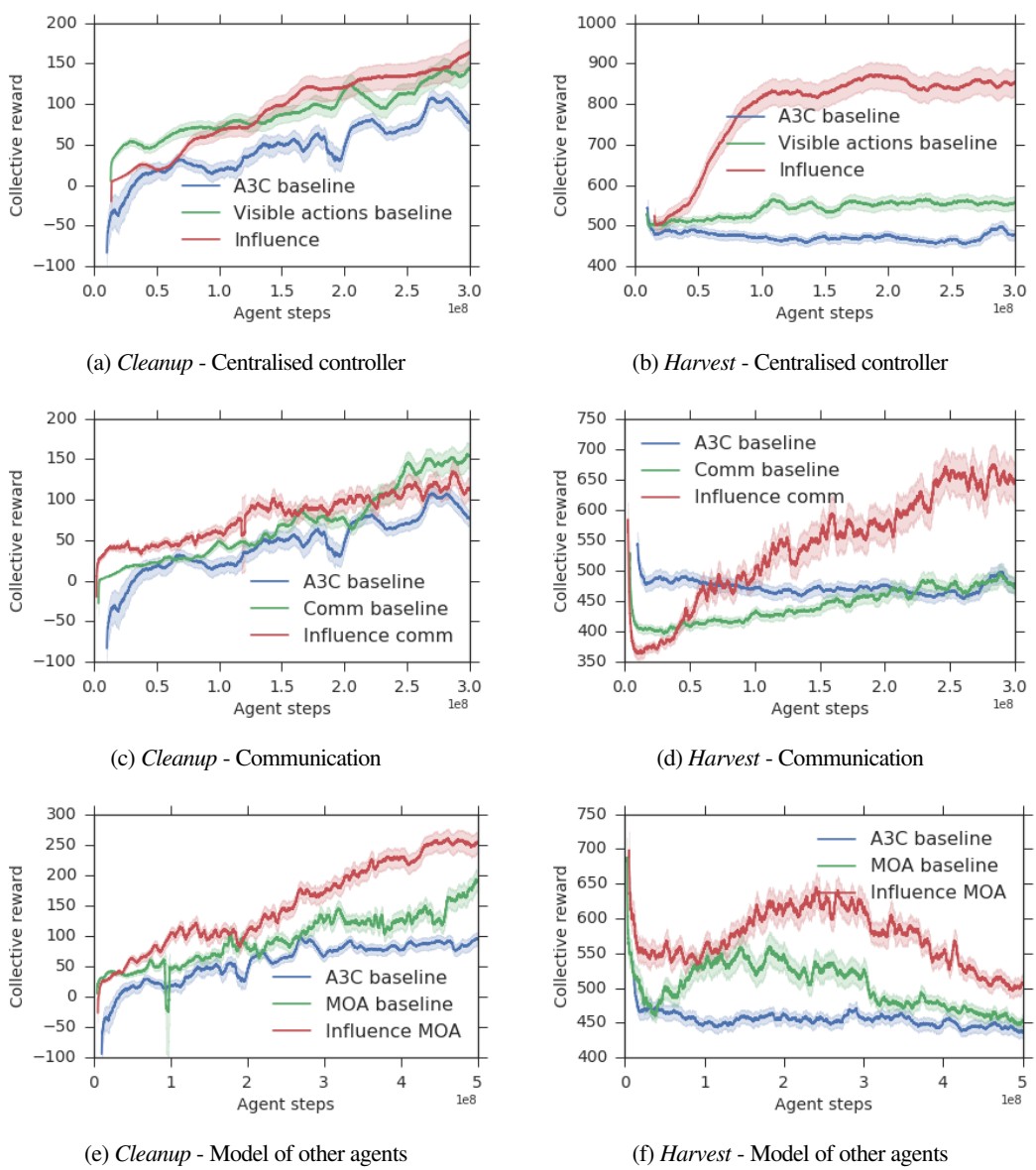

Figure 14: Total collective reward over the top 5 hyperparameter settings, with 5 random seeds each, for all experiments. Error bars show a 99.5% confidence interval (CI) computed within a sliding window of 200 agent steps. The influence models still maintain an advantage over the baselines and ablated models, suggesting the technique is robust to the hyperparameter settings.

### 6.4.1 Optimizing for collective reward

In this section we include the results of training explicitly prosocial agents, which directly optimize for the collective reward of all agents. Previous work (e.g. Peysakhovich & Lerer (2018)) has shown that training agents to optimize for the rewards of other agents can help the group to obtain better collective outcomes. Following a similar principle, we implemented agents that optimize for a convex combination of their own individual reward $E^k$ and the collective reward of all other agents, $\sum_{i=1,i\neq k}^{N} E^i$. Thus, the reward function for agent $k$ is $R^k = E^k + \eta \sum_{i=1,i\neq k}^{N} E^i$. We conducted the same hyperparameter search over the parameters mentioned in Section 6.3.1 varying the weight placed on the collective reward, $\eta \in [0,2]$.

As expected, we find that agents trained to optimize for collective reward attain higher collective reward in both *Cleanup* and *Harvest*, as is shown in Figure 15. In both games, the optimal value for $\eta = 0.85$.

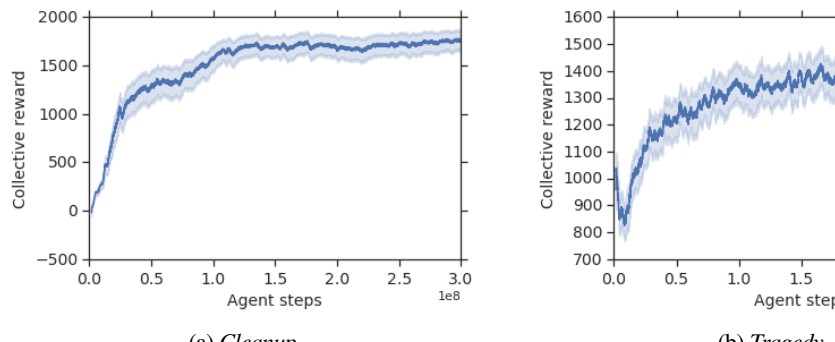

(a) *Cleanup*                                          (b) *Tragedy*

Figure 15: Total collective reward obtained by agents trained to optimize for the collective reward, for the 5 best hyperparameter settings with 5 random seeds each. Error bars show a 99.5% confidence interval (CI) computed within a sliding window of 200 agent steps.

Interestingly, however, the equality in the individual returns for these agents is extremely low. Across the hyperparameter sweep, no solution to the *Cleanup* game which scored more than 20 points in terms of collective return was found in which all agents scored an individual return above 0. It seems that in *Cleanup*, when agents are trained to optimize for collective return, they converge on a solution in which some agents never receive any reward.

Note that training agents to optimize for collective reward requires that each agent can view the rewards obtained by other agents. As discussed previously, the social influence reward is a novel way to obtain cooperative behavior, that does not require making this assumption.

