# OpenReview forum: "Intrinsic Social Motivation via Causal Influence in Multi-Agent RL"
_ICLR.cc/2019/Conference_

### Official Review · AnonReviewer2 · 2018-11-03
**Review of Paper "Intrinsic Social Motivation via Causal Influence in Multi-Agent RL"**

**Rating:** 6
**Confidence:** 3

**Review:**

This paper proposes an approach to model social influence in a scenario-independent manner by instantiating the concept of intrinsic motivation and combine various human abilities as part of a reinforcement learning function in order to improve the agent's operation in social dilemma scenarios.

Agents are operationalised as convolutional neural network, linear layers and LSTM. Using these base mechanisms, different abilities (communication, models of other agents (MOA)), their causal influence is inferred based on counterfactual actions. The architecture is explored across two different sequential social dilemmas.

The architecture is described in sufficient detail, with particular focus on the isolation of causal influence for communication and MOA influence. The experimental evaluation is described in sufficient detail, given the low complexity of the scenarios. While the agents with communicative ability and MOA show superior performance, a few results warrant clarification.

Figure 6a) highlights the performance of influencers in contrast to a visible actions baseline. This specific scenarios shows the necessity to run experiments for larger number of runs, since it appears that action observations may actually outperform influencer performance beyond 3 steps. Please clarify what is happening in this specific case, and secondly, justify your choice of steps used in the experimental evaluation.

Another results that requires clarification is Figure 6f), which is not sufficiently discussed in the text, yet provides interesting patterns between the MOA baseline performance decaying abruptly at around 3 steps, with the influence MOA variant only peaking after that. Please clarify the observation. Also, could you draw conclusions or directions for a combination of the different approaches to maximise the performance (more generally, beyond this specific observation)?

A valuable discussion is the exemplification of specific agent behaviour on Page 7. While it clarifies the signalling of resources in this specific case, it also shows shortcomings of the model's realism. How would the model perform if agents had limited resources and would die upon depletion (e.g. the de facto altruistic influencer in this scenario - since it only performs two distinct actions)? The extent of generalisability should be considered in the discussion.

In general, the paper motivates and discusses the underlying work in great detail and is written in an accessible manner (minor comment: the acronym LSTM is not explicitly introduced). The quality of presentation is good.

---

> ### Author Response · Authors · 2018-11-10
> **Explanation and discussion**
>
> Thanks for your questions about the results in Figure 6. With regards to Figure 6a, we set the limit of 3e8 steps for the first two experiments a priori, and did not change it based on the results of the experiments, to ensure we did not bias the results. While it does appear that the visible actions baseline may reach the performance of the influence model in this experiment, we consider the initial centralized controller experiments to be a simple proof-of-concept. In practice, one would most likely always prefer to use the MOA method for computing influence, since it provides the important benefit that agents do not need to observe each others’ reward or require training with a centralized controller in order to compute influence. As is evident in Figures 6c, 13e, and 14e, the MOA method reliably and clearly outperforms all baselines in the Cleanup game.
>
> With regard to Figure 6f, because multi-agent training is highly stochastic and non-stationary, as agents learn to adapt to each other it can change the dynamics of the environment such that formerly effective strategies no longer result in high reward. For example, in Harvest, as agents get more proficient at collecting apples efficiently, they may actually deplete the apples faster, thus paradoxically lowering overall reward.  As noted in Section 6.4 of the Appendix, if one agent fails to learn to collect apples, it actually makes Harvest easier for the other agents, since the apples are less easily exhausted. However, if this agent then begins to collect apples they will quickly be exhausted. Figure 6f shows some of these unstable dynamics for a single hyperparameter setting with 5 random seeds in Harvest. However, Figure 14f in the Appendix plots the same game using 5 hyperparameter settings with 5 seeds each, giving a more stable training curve.
>
> You make an excellent point about the fact that agents must balance their own self-interest with the intrinsic reward of influencing others. We actually hypothesize that the reason the agent in the example on page 7 learned to communicate was because communication is the cheapest way to obtain influence while still pursuing its own environment reward. In terms of generalizing to new tasks, it is straightforward to tune the parameter which trades off the environment and influence rewards to suit a new task. We will add further discussion about this trade off to in an updated version of the paper.
>
> We have also added text introducing the acronym LSTM - thanks for pointing that out.

---

### Official Review · AnonReviewer1 · 2018-11-05

**Rating:** 4
**Confidence:** 5

**Review:**

INTRINSIC SOCIAL MOTIVATION VIA CAUSAL INFLUENCE IN MULTI-AGENT RL

Main Idea: The authors consider adding a reward term to standard MARL which is the mutual information between its actions and the actions of others. They show that adding this intrinsic social motivation can lead to increased cooperation in several social dilemmas.

Strong Points:
-	This paper is a novel extension of ideas from single agent RL to multi agent RL, there are clear benefits from doing reward shaping in the right way to make deep RL work better.
-	The paper focuses on cooperative environments which is an underfocused area in RL right now

Weak Points:
-	There is missing discussion of a lot of literature. The causal influence term can be thought of as a form of reward shaping. There is little discussion on the (large) literature on reward shaping to get MARL to exhibit good behavior.
-	The results feel quite thin. Related to the point above: the theory of different types of reward shaping (e.g. optimistic Q-learning, prosociality, etc…) are well understood. It is not clear to me under what conditions the authors’ proposed augmentation to the reward function will lead to better or worse outcomes. The experiments in this paper are quite simple and only span a small set of environments so it would be good to have at least some formal theory.
-	Social dilemmas don’t seem like the best application. The authors define the social dilemma as: “For each individual agent, ‘defecting’ i.e. non-cooperative behavior has the highest payoff.” With the intrinsic motivation, agents learn to cooperate. This is good, however, if we’re thinking about situations where agents aren’t trained together and have their own rewards (the authors’ example: “autonomous vehicles are likely to be produced by a wide variety of organizations and institutions with mixed motivations”) then won’t these agents be exploited by rational agents? Other solutions to this problem (e.g. recent papers on tit-for-tat by Lerer & Peysakhovich or LOLA by Foerster et al. construct agents where defectors get explicitly punished and so don’t want to try exploiting). Is there something I am missing here? Do the agents learn to punish non-cooperators (if no, isn’t it rational at that point to just not cooperate and won’t self-driving cars trained via this method get exploited by others)?
-	Relate to the point(s) above: a better environment for application here seems to be coordination games/”Stag Hunt” games where it is known that MARL converges to poor equilibria and many other methods e.g. optimistic Q-learning or prosociality have been invented to make things work better. Perhaps the method proposed here will work better than these (and it has the appealing property that it does not require the ability to observe the other agents' rewards as e.g. prosociality does)
-	This paper contains some quite grandiose language connecting the proposed reward shaping to “how humans learn” (example: It may also have correlates in human cognition; experiments show that newborn infants are sensitive to correspondences between their own actions and the actions of other people, and use this to coordinate their behavior with others) it’s unclear to me that humans experience extra reward for their actions having high mutual information (and/or causal information with others). While it’s fine to argue some of these points at a high level I would suggest scrubbing the text of the gratuitous references to this.

Nits:
“Crawford & Sobel (1982) find that once agents’ interests diverge by a finite amount, no communication is to be expected.” – this is an awkward phrasing of the Crawford and Sobel result (it can be read as “if interests diverge by any epsilon there can be no communication”). The CS result is that information revealed in communication (in equilibrium) is proportional to amount of common interest.

---

> ### Author Response · Authors · 2018-11-10
> **Explanation of the results and contributions and summary of changes (1/2)**
>
> Thank you for your detailed feedback on our paper, we appreciate your insight. You are right that the reference to human cognition may be a bit grandiose, and we have uploaded a revised version of the paper in which we have rephrased this reference to show that we are only deriving a loose inspiration from this work. We are also happy to remove it entirely if you think it detracts rather than adds from the paper. We have also rephrased the reference to Crawford and Sobel as you suggested.
>
> In addition to the contribution correctly pointed out in your introductory paragraph, the paper also presents results on emergent communication. Further, it shows that the proposed causal influence reward, with estimated models of other agents, enables decentralized training of deep recurrent neural network agents in very challenging settings: directly from pixels, subject to partial-observability and with no knowledge of other agents’ rewards. Previous deep MARL works, e.g. the emergent communication works of Foerster et al., resorted to centralized training because their decentralized approaches failed to learn. This paper innovates a solution to this learning problem, and as such we feel it makes an important contribution.
>
> To your point about comparing the results to those obtained via prosocial reward functions, we actually do compare to a recent method that gives agents a prosocial inductive bias (Inequity Aversion [1]) in Table 4 of the Appendix. These results show that our method is able to exceed the performance of Inequity Aversion in several experiments, in spite of the fact that agents trained with influence do not have access to other agents’ rewards, while prosocial agents do, as you correctly point out. We can also compute the results of comparing to agents trained to optimize directly for the group reward, if you think this is important to include. However, we note that explicitly programming pro-sociality into agents by training them to optimize for group reward, as in [3], can lead to problems with spurious rewards and “lazy agents”, and is not possible if the reward functions of other agents are unknown. Influence is a more general approach, and as such it represents a significant contribution to the state-of-the-art.
>
> We agree that testing on Stag Hunt would be an interesting future extension, and expect that our method would likely perform well in this game, since emergent communication should allow agents to coordinate better and thus be beneficial for all agents. We are also excited about testing whether influence can improve coordination in manipulation and control tasks as well. The reason we initially focused on the Tragedy of the Commons and Public Goods games presented in this paper is because we felt that they would present the most difficult challenge, since they not only require coordination, but also require that coordination to be prosocial. These were known to be the hardest established benchmark tasks in this domain, and allowed us to compare easily with prior work [1-2]. It is known that vanilla MARL converges to poor equilibria in these games as well. Since this paper is mainly about presenting a new agent algorithm, we felt that running comparisons on all possible kinds of games from game theory would be out of scope, especially given length restrictions on the paper; we decided that presenting results indicating that influence can be used to train effective communication protocols was a more important contribution.
>
> Because of the points above, we must respectfully disagree with your assertion that the results are thin. We give results from 3 experiments which are stable across 15 hyper-parameters, each with 5 seeds, and using multiple metrics (please see the Appendix for additional results, including further results testing on a 3rd, proof-of-concept environment). These results constitute very strong empirical evidence obtained in a rigorous way. Our experiments involve multiple agents with memory acting under partial observability, and with non-trivial, nonlinear, high-dimensional, recurrent policies. These aspects of the problem make it very challenging from an empirical perspective, and beyond theoretical analysis using existing tools.

---

> ### Author Response · Authors · 2018-11-10
> **Explanation of the results and contributions and summary of changes (2/2)**
>
> Thank you for pointing out the connection to related work on reward shaping. We initially understood reward shaping to be specific to a given environment, and would argue that intrinsic motivation is designed to be a more general mechanism that works across environments, and thus focused on related work in intrinsic motivation. However, at your suggestion we have begun looking for related work in the reward shaping literature (such as [4-5]) and after reading these works in detail, will include references to them in an updated version of the text. We are happy to include other specific papers that you can recommend.
>
> You raise an interesting question about whether the influence reward, if used to train autonomous vehicles, could lead to vehicles being exploited for information. The example of autonomous driving was mainly meant to illustrate the benefit of decentralized training. Obviously the problem of cars driving in the real world is much more complex than the simulations tested here, and so we cannot make claims about whether the influence reward could generalize to this setting. However, it is interesting to consider the question of the degree to which the desire to influence may lead to being exploited. Since the agents balance both influence and environmental reward based on a hyperparameter, this can be tuned to ensure influence does not override the drive for environmental reward. We hypothesize that sharing information is actually a relatively cheap way to influence another agent, without sacrificing much in terms of one’s own environmental reward; this may protect agents from being unduly exploited. However, we should emphasize that we think the approach of training agents with influence goes well beyond the application of autonomous vehicles. As we have shown in the paper, influence can be an effective way to train agents to learn to communicate with each other, and could thus be valuable whenever meaningful communication is desired. We think that this could be an important and novel contribution to the emergent communication community.
>
> [1] Edward Hughes, Joel Z Leibo, Matthew G Phillips, Karl Tuyls, Edgar A Duenez-Guzman, Antonio Garcıa Castaneda, Iain Dunning, Tina Zhu, Kevin R McKee, Raphael Koster, et al. Inequity aversion improves cooperation in intertemporal social dilemmas. In Advances in neural information processing systems (NIPS), Montreal, Canada, 2018.
>
> [2] Joel Z Leibo, Vinicius Zambaldi, Marc Lanctot, Janusz Marecki, and Thore Graepel. Multi-agent reinforcement learning in sequential social dilemmas. In Proceedings of the 16th Conference on Autonomous Agents and MultiAgent Systems, pp. 464–473. International Foundation for Autonomous Agents and Multiagent Systems, 2017.
>
> [3] Devlin, S., Yliniemi, L., Kudenko, D., & Tumer, K. (2014, May). Potential-based difference rewards for multiagent reinforcement learning. In Proceedings of the 2014 international conference on Autonomous agents and multi-agent systems (pp. 165-172). International Foundation for Autonomous Agents and Multiagent Systems.
>
> [4] Devlin, S., Yliniemi, L., Kudenko, D., & Tumer, K. (2014, May). Potential-based difference rewards for multiagent reinforcement learning. In Proceedings of the 2014 international conference on Autonomous agents and multi-agent systems (pp. 165-172). International Foundation for Autonomous Agents and Multiagent Systems.
>
> [5] Peysakhovich, A., & Lerer, A. (2018, July). Prosocial learning agents solve generalized stag hunts better than selfish ones. In Proceedings of the 17th International Conference on Autonomous Agents and MultiAgent Systems (pp. 2043-2044). International Foundation for Autonomous Agents and Multiagent Systems.

---

### Official Review · AnonReviewer3 · 2018-11-12
**Interesting paper, possibly some confusion on some causal modelling (especially Section 2.1)**

**Rating:** 5
**Confidence:** 3

**Review:**

The paper introduces a new intrinsic reward for MARL, representing the causal influence of an agent’s action on another agent counterfactually. The authors show this causal influence reward is related to maximising the mutual information between the agents’ actions. The behaviour of agents using this reward is tested in a set of social dilemmas, where it leads to increased cooperation and communication protocols, especially if given an explicit communication channel. As opposed to related work, the authors also equip the agents with an internal Model of Other Agents that predicts the actions of other agents and simulates counterfactuals. This allows the method to run in a decentralized fashion and without access to other agents’ reward functions.

The paper proposes a very interesting approach. I’m not a MARL expert, so I focused more on the the causal aspects. The paper seems generally well-organized and well-written, although I’m a bit confused about the some of the causal modelling decisions and assumptions. This confusion and  some potential errors, which I describe in detail below, are the reason for my borderline decision, despite liking the paper otherwise.

First, I’m a bit confused about the utility of the Section 2.1 model (Figure 1), mostly because of the temporal and multiple agents aspects that seem to be dealt with (“more”) correctly in the MOA model. Specifically in Figure 1, one would need to assume that there is only one agent A influencing agent B at the same time (and agent B does not influence anything else). For example, there is no other agent C which actions also influence agent B, and no agent D that is influenced by agent B, otherwise the backdoor-criterion would not work, unless you add also the action of agent C to the conditioning set (or its state). Importantly, adding the actions of all agents, also a potential agent D that is downstream of B would be incorrect. So in this model there is some kind of same time interaction and there seems to be the need for a causal graph that is known a priori. These problems should disappear if one assumes that only the time t-1 actions can influence the time t actions, as in the MOA model. I assume the idea of the Figure 1 model was to show a relationship with mutual information, but for me specifically it was quite confusing.

I was much less confused by the MOA causal graph represented in Figure 4, although I suspect there are quite some interactions missing (for example s_t^A causes u_t^A similarly to the green background? s_t causes s_{t+1} (which btw in this case should probably be split in two nodes, one s_{t+1} and one s_{t+1}^B?). Possibly one could also add the previous time step for agent B (with u_{t+1}^B influenced by u_t^B I would assume?). As far as I can see there is no need to condition on a_t^B in this case to see the influence of a_t^A on a_{t+1}^B, u_t^A and s_t^A should be enough?

Minor details:
Is there possibly a log missing in Eq. 2?

---

> ### Author Response · Authors · 2018-11-19
> **Clarifications on causal modeling**
>
> Thanks for your feedback - we are glad that you found the paper interesting, and we hope to be able to clear up any confusion surrounding the causal modeling.
>
> You are correct that the first method of implementing the causal influence reward described in section 2.1 has the important limitation that agents cannot mutually influence each other. However, we believe we have handled the conditioning correctly to satisfy the back door criterion, by imposing a sequential ordering on agents’ actions. We allow only a fixed number of agents to be influencers, and the rest are influencees. Only an influencer gets the causal influence reward, and only an influencee can be influenced. At each timestep, the influencers choose their actions first, and these actions are then given as input to the influencees. Let’s say that agent A and B are influencers, and C is an influencee. Then C receives both a^A_t and a^B_t as input. When computing the causal influence of A on C, we also add a^B_t to the conditioning set, as you describe. However, we do not condition on actions downstream of C, as you mention. You are correct that in this model the causal graph does need to be known a priori, and in that sense it is more limited. We only introduced this initial model as a proof-of-concept, and retained it in the paper because it is associated with some of the interesting qualitative results we present in Section 4.1. We will modify the paper to include a more detailed description of the sequential nature of agents’ actions in order to reduce confusion in the future. However, you are correct that the MOA approach is likely to be more effective in practice, and we would like to emphasize the success of this approach, and the communication results in Section 4.2, as more important contributions.
>
> You are right that we are missing an arrow from s_t -> s_{t+1}, and the partially observed states s^B_{t+1} in Figure 4; we will add these to the Figure and update it in the next revision. You are also correct that we do not need to condition on a_t^B, but we do allow the model to use a_t^B when making its predictions about a_{t+1}^B, so we have shown this as shaded in the Figure.
>
> We don’t believe there is a missing log in equation 2; the log is absorbed into the KL term.

---

### Author Response · Authors · 2018-11-21
**Summary of revisions**

In response to the feedback provided by the reviewers, we have made the following revisions to the paper:

- Added Section 6.4.1 to the Appendix, which provides additional results from training prosocial agents to directly optimize for group reward.

- Added further explanation on the causal inference in the centralized controller case to Section 2.1.

- Revised the MOA causal graph in Figure 4.

- Added the following references to the related work: Peysakhovich & Lerer (2018), Devlin et al. (2014), Foerster et al. (2018), Oudeyer & Kaplan (2006), Oudeyer & Smith (2016), Forestier & Oudeyer (2017).

- Added a future work section suggesting further applications of the influence reward.

- Rephrased the reference to Crawford & Sobel (1982) in the related work.

- Modified language in the introduction citing Tomasello’s research on infant cognition.

- Corrected references to “Table 11” to Table 4.

- Introduced LSTM acronym.

---

### Public Comment · ~Rachit_Dubey1 · 2018-11-21
**Interesting work and comments on the introduction**

I found this paper to be really interesting and wanted to congratulate the authors on a neat idea that is very well executed. Good luck with the reviews.

That being said, as a cognitive scientist I find the claims made in the introduction to be slightly misrepresentative of the field of cognitive science and psychology and would suggest/request for a slight modification of those statements. Specifically, the statement "Arguably, the most extraordinary aspect of human intelligence is not curiosity or a drive for power; rather it is our remarkable social abilities which have given rise to cultural evolution, and unprecedented progress and coordination on a massive scale" is too strong with no real empirical and theoretical evidence behind the same. The human mind is very complex and I would be skeptical of reducing intelligence to simply one factor. Furthermore, I don't think cognitive science has advanced enough to the point to say what is the most critical aspect of human cognition (this is still widely debated and much work remains to be done to understand this).

With this in mind, here is my suggestion - I don't think the paper would be drastically affected if the authors were to point how critical social abilities are for human intelligence and then use that as a motivation for their work (as opposed to claiming that to be the most important aspect of cognition which remains unproven). I also want to state my reason for this suggestion/request  - I can see this paper to be widely cited in the future and I think more and more papers in computer science will continue to draw inspiration from psychology. Therefore, it is also important that those psychology works are properly explained so that computer scientists not only appreciate the beauty of the human mind but are also aware of the complexity of our fascinating mind for continued inspiration.

Thanks a lot and congrats again on a great paper.

---

> ### Author Response · Authors · 2018-11-21
> **Thanks for the valuable input!**
>
> Thank you so much for your interest in the paper! It’s wonderful to hear that you think it may be well-cited in the future.
>
> We truly value your insight as a cognitive scientist and would be happy to revise the intro according to your suggestions. What do you think of the following revised language:
>
> “Humans have remarkable social learning abilities; some authors suggest that it is social learning that has given rise to cultural evolution, and allowed us to achieve unprecedented progress and coordination on a massive scale \citep{van2011social, Herrmann1360}. Others emphasize that our impressive capacity to learn from others far surpasses that of other animals, apes, and even other proto-human species \citep{henrich2015, harari2014sapiens, laland2017}.”

---

> > ### Public Comment · ~Rachit_Dubey1 · 2018-11-22
> > **Thanks for your reply**
> >
> > I am glad that you liked and appreciated my suggestion. The rephrased paragraph looks great - thanks for modifying that and good luck with the final reviews!

---

### Author Response · Authors · 2018-12-20
**No response to revisions**

We would like to point out that the reviewers have not posted a response to our revisions. We replied to all concerns listed by the reviewers and made extensive revisions to the paper. Further, two of the reviewers appeared to like the paper (saying that the quality is good, that it is interesting, that they liked it), but only asked for clarifications. We provided these clarifications, but the reviewers have not acknowledged our response / revisions nor updated their scores.

---

### Meta-Review · Area_Chair1 · 2018-12-19

**Confidence:** 4
**Recommendation:** Reject

**Metareview:**

The reviewers raised a number of concerns including the appropriateness of the chosen application and the terms in which social dilemmas have been discussed, the lack of explanations and discussions, missing references, and the extent of the evaluation studies. The authors’ rebuttal addressed some of the reviewers’ concerns but not fully. Overall, I believe that the work is interesting and may be useful to the community (though to a small extent., in my opinion). However, the paper would benefit from additional explanations, experiments and discussions pointed in quite some detail by the reviewers. AS is, the paper is below the acceptance threshold for presentation at ICLR.